# Febrile Seizures Cause Depression and Anxiogenic Behaviors in Rats

**DOI:** 10.3390/cells11203228

**Published:** 2022-10-14

**Authors:** Yeon Hee Yu, Seong-Wook Kim, Hyuna Im, Yejin Song, Seo Jeong Kim, Yu Ran Lee, Gun Woo Kim, Changmin Hwang, Dae-Kyoon Park, Duk-Soo Kim

**Affiliations:** 1Department of Anatomy, College of Medicine, Soonchunhyang University, Cheonan-si 31151, Korea; 2BK21 FOUR Project, College of Medicine, Soonchunhyang University, Cheonan-si 31151, Korea; 3Graduate School of New Drug Discovery & Development, Chungnam National University, 99 Daehak-ro, Yuseong-gu, Daejeon 34134, Korea

**Keywords:** febrile seizure, emotional phenotypes, hippocampus, local field potentials, theta oscillations

## Abstract

Febrile seizure (FS) is a common type of seizure occurring in human during infancy and childhood. Although an epileptic seizure is associated with psychiatric disorders and comorbid diseases such as depression, anxiety, autism spectrum disorders, sleep disorders, attention deficits, cognitive impairment, and migraine, the causal relationship between FS and psychiatric disorders is poorly understood. The objective of the current study was to investigate the relationship of FS occurrence in childhood with the pathogenesis of anxiety disorder and depression using an FS rat model. We induced febrile seizures in infantile rats (11 days postnatal) using a mercury vapor lamp. At 3 weeks and 12 weeks after FS induction, we examined behaviors and recorded local field potentials (LFPs) to assess anxiety and depression disorder. Interestingly, after FS induction in infantile rats, anxiogenic behaviors and depression-like phenotypes were found in both adult and juvenile FS rats. The analysis of LFPs revealed that 4–7 Hz hippocampal theta rhythm, a neural oscillatory marker for anxiety disorder, was significantly increased in FS rats compared with their wild-type littermates. Taken together, our findings suggest that FS occurrence in infants is causally related to increased levels of anxiety-related behaviors and depression-like symptoms in juvenile and adult rodents.

## 1. Introduction

Febrile seizure (FS) is a type of seizure induced by fever in infants and young children. It occurs in 3–5% of children between 6 months and 5 years of age [1]. FS occurs as a result of fever above about 38 °C due to non-central nervous system (CNS) infections [2]. Fever is one of the oldest biomarkers. It is associated with changes in neuronal activity in the CNS [3,4]. Hyperthermia can be induced by the excitability of principal cells in the hippocampus [2]. A hyperthermic seizure for only 20 min can cause acute hippocampal neuronal injury [1]. FS in humans does not lead to behavioral deficits later for life, although epileptic seizure can increase the risk of neurological disorders [5]. However, many animal studies have shown that prolonged FS might be linked to alterations in hippocampal neural oscillations [1,6,7].

On the other hand, anxiety and depressive symptoms are among the most frequent aftereffects of seizure that can negatively affect the quality of life of people with epilepsy. Emotional behaviors are different depending on the epileptic condition, diverse status epilepticus (SE) models, and rat strains used in experimental temporal lobe epilepsy (TLE) [8]. Although affective disorders are common comorbidities of TLE, their underlying mechanisms are poorly understood. Since affective symptoms are known to be associated with secondary seizures, researchers have hypothesized that some biological factors of seizures might lead to affective disturbances [9]. Another study has suggested that the long-term cognitive impact of a single early-life seizure induced by kainate is largely limited to the hippocampus and prefrontal cortex [10]. However, in SE models, neuronal loss is not restricted to the hippocampus. It also occurs in the parahippocampal cortical regions, the thalamus, the endopiriform cortex, and the amygdala [10,11,12,13]. Among these areas, the amygdala is involved in cognitive deficits and emotional disorders in humans and other animals [14,15].

Although the causal relationship between FS occurrence and emotional disorders [16,17,18] remains controversial, previous studies have clearly indicated that early developmental insults by FS can aggravate medial temporal lobe functions associated with facial emotional expression and recognition in human [19], as well as cause anxiety-related behaviors in adult rodents [19]. However, most of the prior studies were focused on epileptogenic activity in adulthood after early-life seizure impairment rather than primarily on behavioral changes. To the best of our knowledge, little data are available pertaining to the relationship between the long-term effect of epilepsy and declined emotional phenotypes after FS. Thus, the objective of the current study was to perform a comparative analysis of emotional phenotype using juvenile and adult rodents with FS to improve our understanding of the behavioral sequelae of FS during the epileptogenic period and adulthood.

## 2. Materials and Methods

### 2.1. Animals

In all experiments, the progeny of Sprague–Dawley (SD) rats obtained from Experimental Animal Center, Soonchunhyang University (Cheonan, South Korea) were used. All animals were provided with a commercial diet and water ad libitum under controlled temperature, humidity, and lighting conditions (light/dark cycle, 12 h:12 h; temperature, 22 ± 2 °C; humidity, 55 ± 5%). All animal protocols were approved by the Administrative Panel on Laboratory Animal Care of Soonchunhyang University (permit No. SCH16-092). All efforts were made to minimize the number of animals used and their suffering.

### 2.2. FS Induction

A hyperthermic seizure rat model of FS has been previously described [7]. Briefly, rat pups (11 days postnatal) were used because their hippocampal developmental stage at this age is generally equivalent to human infants [20,21]. After punch-marking of their ears, the pups were warmed in a plastic chamber measuring 10 cm × 13 cm at the base and 12 cm in height. The chamber floor was covered with paper towels. A 175 W mercury vapor lamp was held 3 cm above the chamber. Core temperatures were measured before inducing hyperthermic seizure and every 5 min during seizure induction using an ear thermoprobe. All rats displayed generalized seizures approximately 5–10 min after hyperthermic seizure induction when the core temperature was 41–43 °C. Hyperthermic seizures were maintained for 40 min after generalized seizure onset because the duration of FS was maintained for at least 15 min with more than one seizure in a day, resulting in transient neuronal injury and epilepsy [22,23]. During maintenance of hyperthermic seizures, rats showed multiple generalized seizures (stages 4 and 5) [24]. After hyperthermic seizure, the rats were moved to a cool surface, and returned to their mothers (mortality of approximately 3%). Siblings placed in a chamber at room temperature were used as a control group. In order to completely identify FS induction, we observed recurrent seizure at an interval of approximately 4 h each day in the vivarium for general behaviors (Racin scale criteria, 2.3 ± 0.07). In addition, we performed a double cross-check by performing LFPs monitoring to examine altered hippocampal oscillations caused by FS induction [6,7]. In this study, only rats with complete FS induction were used. After recurrent seizure onset (10–12 weeks after hyperthermic seizure induction, ~71% approximately), behavioral seizures were scored based on Racine scale criteria (stage 1, mouth and facial movements; stage 2, head nodding; stage 3, forelimb clonus; stage 4, rearing; stage 5, rearing and falling) [24].

### 2.3. Behavioral Tests

All rats were tested for behavioral traits at 3 weeks after FS induction (FS 3 weeks) and 12 weeks after FS induction (FS 12 weeks). Behavioral tests were recorded and analyzed using a PC-based video behavior analysis system with automated tracking software Noldus EthoVision 3.1 (Noldus, Leesburg, VA, USA).

#### 2.3.1. Open-Field Test

The open-field arena was designed for FS 3 weeks (40 cm × 40 cm × 40 cm) and FS 12 weeks (60 cm × 60 cm × 40 cm), respectively. It was divided into a central zone (20 cm × 20 cm, 30 cm × 30 cm; each for FS 3 weeks and FS 12 weeks) and a border zone. Rats were placed in the center zone of the open-field arena and allowed to explore freely for 30 min. Total distances moved and the time spent in the center zone were measured to evaluate locomotor activity and anxiety levels, respectively [25,26].

#### 2.3.2. Light–Dark Transition Test

The light–dark transition apparatus was composed of a white compartment (light box, 30 cm × 30 cm × 30 cm) and a black-enclosed compartment (dark box, 30 cm × 20 cm × 30 cm). The light box was brightly illuminated (300 lux). Rats were placed in the center zone of the light box and allowed to move freely for 5 min. Total time spent in the light box was measured to evaluate the anxiety levels of animals [26,27].

#### 2.3.3. Zero Maze Test

The zero maze apparatus made of polyethylene was located 50 cm above the floor. It consisted of two open arms (10 cm in width and 50 cm in length) connected perpendicular to two closed arms of equal dimensions with a 10 cm square center region. The two closed arms included black walls of 30 cm in height. Curbs (1 cm) were installed along edges to prevent falls from open arms [28]. Rats were put in the center region facing a closed arm and allowed to explore for 5 min. The number of entries into individual arms and the time spent on individual arms were measured for 5 min. Time spent in open arms was assessed as an index of anti-anxiety behavior.

#### 2.3.4. Elevated plus Maze Test

Elevated plus maze test was performed as described previously [26]. The elevated-plus maze apparatus consisted of four arms elevated 60 cm above the floor. The length and width of each arm were 50 cm and 10 cm, respectively. Two of the arms were opened while the others were closed by 50 cm high walls. Rats were placed at the central plate and allowed to move freely for 5 min [26]. The number of entries into individual arms and the time spent on individual arms were measured for 5 min. Rats were considered to be in the open arms only if all four paws were in the open arm of the maze.

#### 2.3.5. Forced Swim Test

Forced swim test was performed to measure depression-like behaviors as described previously [29,30]. Rats were individually placed into a plastic cylinder (50 cm height, 30 cm in diameter) filled with water (23 ± 3 °C). Behavior was monitored for 5 min, and immobility time was measured. The time spent immobile was considered to reflect depression-like behaviors. After the session, rats were removed from the pool, dried with a towel, and returned into their cages.

### 2.4. Local Field Potentials (LFPs)

At each designated time point (3 weeks and 12 weeks) in the wild-type group and FS group, animals were anesthetized by intraperitoneal injection of urethane (1.5 g/kg) and placed in a stereotaxic frame. Holes were drilled through the skull to introduce the electrodes. For juvenile rats (3 weeks after FS), LFPs were recorded using a tungsten parylene electrode (0.005-inch outer diameter; A-M Systems, Sequim, WA, USA). Coordinates were as follows (relative to CA1): 3.5 mm posterior to bregma, 2.0 mm lateral to midline, and 2.4 mm depth. For adult rats (12 weeks after FS), glass microelectrodes (microfilament capillary 1.2 outer diameter; 5–10 MΩ) filled with artificial cerebrospinal fluid (ACSF, in mM; NaCl 126, KCl_5_, CaCl_2_ 2, MgCl_2_ 2, NaH_2_PO_4_ 1.25, NaHCO_3_ 26, D-glucose 10, pH 7.2) were used. Coordinates were as follows (relative to CA1): 3.8 mm posterior to bregma, 2.5 mm lateral to midline, and 2.9 mm depth. LFPs were recorded with a QP511 AC amplifier (0.1–3000 Hz bandpass, GRASS Technologies, West Warwick, RI, USA). Data were digitized (5 kHz) and recorded to obtain baseline values for 2 h. Single-channel acquisition was performed using an Axoscope 10.2 software (Axon Instruments, San Jose, CA, USA). Analysis of single-channel electrical traces was conducted using a Clampfit 10.2 software (Axon Instruments, Burlingame, CA, USA). To analyze changes in normalized power of LFP, the amplitude spectrum of normalized power was estimated in event frequency and root mean square (RMS) values were used to derive estimates of spectral power (mV^2^) in 1 Hz frequency bins for each electrode site. Spectral power values were averaged across all epochs within a single baseline. The resulting power was expressed as mV^2^/Hz. For each subject, fast Fourier transform (FFT) of epochs with a resolution of 0.61 Hz was computed for all electrodes and then averaged. Non-overlapping hamming windows of wild-type would lead to spectral leakage. Moreover, FFT power value measurements within each frequency between 1 and 50 Hz were averaged to create 50 non-overlapping <1 Hz frequency bins because frequency bands of interest were defined as: δ (1–4 Hz), θ (4–7 Hz), α (7–12 Hz), and ß (12–25 Hz) [31,32].

### 2.5. Statistical Analysis

All data were analyzed using Student’s *t*-test to determine statistical significance. A *p*-value of <0.05, <0.01, or <0.001 was considered statistically significant.

## 3. Results

### 3.1. Increased Levels of Anxiety-Related Behaviors in FS Rat Model

To determine whether FS occurrences affected anxiety levels in juvenile and adults, we assessed the anxiety-related behaviors of FS 3 weeks and FS 12 weeks rats by performing elevated plus maze test, light–dark transition test, elevated zero maze test, and open-field test. In the elevated plus maze test, the percentages of entries into the open arms of both FS 3 weeks (*p* < 0.001; Student’s *t* test; Figure 1A) and FS 12 weeks rats (*p* < 0.001; Student’s *t*-test; Figure 1E) were significantly decreased in comparison with those of non-FS induced wild-type littermates. In the light–dark transition test, FS 3 weeks rats made fewer transitions from the dark to the light compartment than wild-type rats at 3 weeks (*p* = 0.001; Student’s *t*-test; Figure 1B). The time spent in light chamber was also reduced in FS 3 weeks rats in comparison with wild-type rats at 3 weeks (*p* < 0.001; Student’s *t*-test; Figure 1C). Consistent with results from FS 3 weeks rats, the number of light/dark transitions was also decreased in FS 12 weeks rats in comparison with that in wild-type rats at 12 weeks (*p* = 0.008; Student’s *t*-test; Figure 1F). The time spent in the light chamber for the FS 12 weeks group was also significantly shorter than that for the wild-type group (*p* < 0.001; Student’s *t*-test; Figure 1G). Similarly, in the elevated zero maze test, the total time spent in open arms was markedly decreased in FS 3 weeks rats in comparison with wild-type rats at 3 weeks (FS 3 weeks, *p* < 0.001; Student’s *t*-test; Figure 1D). In the open-field test, the time spent in the center area in the FS 12 weeks group was remarkably reduced in comparison with that in the wild-type group at 12 weeks (FS 12 weeks, *p* < 0.001; Student’s *t*-test; Figure 1H). Taken together, these results show a higher level of anxiety in FS 3 weeks and FS 12 weeks rats than in wild-type littermates.

### 3.2. Increased Levels of Depression in FS Rat Model

Next, to determine whether FS occurrences in rats also caused depression-like phenotypes in juvenile and/or adults, we performed the forced swim test. When we assessed depression-like phenotypes of FS 3 weeks and FS 12 weeks rats, each group showed an increased frequency of immobility (FS 3 weeks, *p* = 0.001; FS 12 weeks, *p* < 0.001; Student’s *t*-test; Figure 2A,D). At 3 weeks and 12 weeks after FS induction in rats, the latency to immobility was significantly decreased in comparison with that of wild-type littermates (FS 3 weeks, *p* = 0.003; FS 12 weeks, *p* < 0.001; Student’s *t*-test; Figure 2B,E). Immobile durations at both time periods in the FS animal groups were markedly elevated in comparison with those of wild-type littermates (FS 3 weeks, *p* < 0.001; FS 12 weeks, *p* < 0.001; Student’s *t*-test; Figure 2C,F). Taken together, our findings indicate that FS can induce depression-like behaviors in rats. Furthermore, these altered emotional phenotypes worsened as the period after FS onset became longer.

### 3.3. Increased Levels of Locomotion in FS Juvenile, but Not Adult, Rat Model

To distinguish characteristics of locomotor activities, an open-field test was performed for wild-type and FS rat models over different epileptogenic time periods. FS 3 weeks rats displayed hyperlocomotion in comparison with wild-type littermates (*p* < 0.001; Student’s *t*-test; Figure 3A,B). FS 12 weeks rats displayed normal levels of locomotor activities, similar to wild-type rats (*p* = 0.22; Student’s *t*-test; Figure 3C,D). Movement tracks (red line) of representative rats in each group are shown in Figure 3A,C.

### 3.4. Representative Profiles of LFP after FS

To examine whether hippocampal 4–12 Hz oscillations were related to emotional states, including anxiety and depression [33], and might be altered in the hippocampus of FS rat models, we recorded LFP in the CA1 region of the rat hippocampus following FS induction. FS 3 weeks rats had similar LFP pattern to wild-type rats of 3 weeks (Figure 4A). However, 1–12 Hz LFP power spectral densities of FS 3 weeks rats were higher than those of wild-type littermates (Figure 4B). A histogram of absolute power quantified at different frequency ranges (1–12 Hz), including delta waves (*p* = 0.016; Student’s *t*-test; Figure 4C), theta waves (*p* < 0.001; Student’s *t*-test; Figure 4D), and alpha waves (*p* = 0.018; Student’s *t*-test; Figure 4E) in FS 3 weeks and wild-type littermates revealed elevated neuronal oscillations at 1–7 Hz in FS 3 weeks rats. FS 3 weeks rats also had higher LFP-normalized power than wild-type littermates (*p* = 0.011; Student’s *t*-test; Figure 4E).

On the other hand, LFP signals of FS 12 weeks rats showed epileptiform discharges characterized by intermittently large amplitude spikes of irregularly sharp waves and multi-spikes in the hippocampal CA1 region in comparison with wild-type 12 weeks rats (Figure 5A). Moreover, power spectral densities of 1–12 Hz LFP were significantly higher in FS 12 weeks rats than in wild-type littermates (Figure 5B). FS 12 weeks rats had higher normalized LFP power than wild-type rats after 12 weeks (*p* = 0.001; Student’s *t*-test; Figure 5C). The neural activity of the delta rhythm is a unique LFP signaling marker of the epileptic hippocampus [34,35]. It was significantly elevated in the hippocampus of FS 12 weeks rats than in wild-type littermates (*p* < 0.001; Student’s *t*-test; Figure 5D). Furthermore, anxiety-related theta waves in FS 12 weeks rats were increased in comparison with those of wild-type littermates (*p* < 0.001; Student’s *t*-test; Figure 5E). Similarly, alpha and beta rhythms associated with synchronous neuronal activity in the CA1 region of hippocampus were markedly upregulated in FS 12 weeks rats compared with wild-type littermates (alpha, *p* < 0.001; beta, *p* < 0.001; Student’s *t*-test; Figure 5F,G).

## 4. Discussion

Emotional behavior changes in FS models are insufficient and ambiguous. Thus, the present study investigated the exact relationship between long-term effects of epileptogenesis and its sequelae of emotional phenotypes following hyperthermic seizure.

Previously, some investigators have suggested that the relationship between anxiety or depressive symptoms and seizures is mediated by physiological stress and directly related to the developing brain [17,36]. In addition, childhood emotional and behavioral problems have higher prevalence in children with epilepsy [17,18]. Moreover, epilepsy resulting from hyperthermia in early life can induce a hyperanxious behavior [18,37,38]. In the present study, we primarily identified behavioral patterns of anxiety and depressive-like symptoms at early time period following FS. On the other hand, previous studies have suggested that SE in neonatal rodents induced by pilocarpine can lead to altered anxiety-related and abnormal social behaviors during adulthood [11,39]. Similar to previous studies, the present study also revealed that emotional phenotypes including anxiogenic and depression increased in the recurrent seizure time period such as adult stage following FS. Meanwhile, asymmetric theta rhythm is a potential biomarker of depression in humans [40]. Theta-frequency synchronization has been observed in human subjects with increased anxiety [41]. Enhanced theta rhythm has been revealed during fear responses and anxiety states, supporting the main role of theta rhythm in modulating these states [42,43,44]. Theta-frequency was associated with anxiogenic and depression characteristics of behavior patterns was notably upregulated during the same developmental time period in the FS animal group than in the wild-type animals in the present study. These phenotypes and brain waves were maintained from the epileptogenic time period to adult stage following FS. The findings of the present study suggest that increased emotional phenotypes and theta oscillation in immature time periods following hyperthermic seizure might be related to the abnormal developmental condition of brain tissues. Because early developmental insults caused by FS can affect medial temporal lobe functions associated with emotional disorders, children with recurrent or prolonged FS might be more defective in neuropsychological tests than typical children or those with simple febrile seizures [16,45]. However, further studies with more subdivided groups are needed to confirm this hypothesis.

Additionally, Ca^2+^ channels among ion channels are implicated in rodent models of comorbid epilepsy and depression [46]. Ca^2+^ channel blockers have been proposed as effective treatment strategies [46]. Increased GABA levels induced by vigabatrin and ethosuximide exhibit both anticonvulsant and antidepressant effects in Wag/Rij rats, supporting the crucial role of GABA in the comorbidity model [47,48]. Thus, GABAergic function may underlie the depressive phenotype of these animals. It has been suggested that depression-related behaviors might involve the role of GABA. Previously, some investigators have reported that the hippocampus is involved in cognitive deficits and emotional disorders in humans and other animals [33,49,50]. Our previous study has identified aberrant inhibitory signaling in the hippocampus of a febrile seizure model [7]. These findings suggest that a long-term deteriorated emotional phenotype, including anxiety and depressive behaviors, might be due to excessive GABAergic functional imbalance caused by abnormalities of Ca^2+^ channels and neurogenesis in the developmental period following a hyperthermic seizure. Unnatural synaptic connections and facilitated abnormal functions in the hippocampus following FS might form a basis for disturbing emotional changes caused by febrile convulsions. However, additional investigations are necessary to test this hypothesis.

The maturation of neurons in the hippocampal circuit is crucial for behavioral abnormalities including working memory deficit and hyperlocomotor activity as well as cognitive function and emotional behavior [51,52,53]. Neonatal lesions in the hippocampus can elicit alterations in the GABAergic system, leading to the functional consequences of brain excitability such as the pathophysiology of schizophrenia [54]. Attention deficit hyperactivity disorder (ADHD, also called hyperkinesia or minimal brain dysfunction) is one of the most common mental disorders among children [55]. Similarly, results of our behavioral experiments showed a massively elevated activity, similar to as shown in ADHD, in a novel environmental state, during the early time period at 3 weeks following FS. According to our previous study, neonatal seizures induced by FS do not lead to cell loss, although they can lead to considerable synaptic reorganization and changes in GABAergic interneurons [7]. Therefore, alterations of locomotor activity in the early period, at 3 weeks following FS, might be accompanied by abnormal immature neuronal populations in the hippocampus and unexpected synaptic reorganization at this developmental stage after a hyperthermic seizure. These alternations induced by FS might affect the prevalence and development of mental disorders including hyperkinesia, ADHD, schizophrenia, and autism during childhood. However, further studies are needed to confirm this hypothesis. On the other hand, in the adult stage following FS, locomotor activities were recovered to levels similar to wild-type levels. These results are consistent with previous reports showing general locomotion following epilepsy in animal models [39,56]. Moreover, the delta rhythm for synchronous neuronal activity in the hippocampus in the 12 weeks FS animal group, which offers a unique LFP signal marker in epilepsy [34,35], was significantly elevated over time. These results suggest that altered LFP signals following FS might result in enhanced epileptiform discharges accompanied by abnormalities of hyperexcitatory and inhibitory neuronal transmission, thus influencing the recurrent seizure period rather than childhood [7].

## 5. Conclusions

In conclusion, our findings suggest that functional imbalances in GABAergic inhibition and abnormalities of unnatural neurogenesis in hippocampal dentate granule cells might lead to time-dependent emotional phenotypes and specific brain waves. In addition, changes in neural oscillations in the hippocampus might be related to cognitive dysfunctions and anxiety disorders, which might be useful as neural oscillatory markers representing depression and emotional disorders.

## Figures and Tables

**Figure 1 cells-11-03228-f001:**
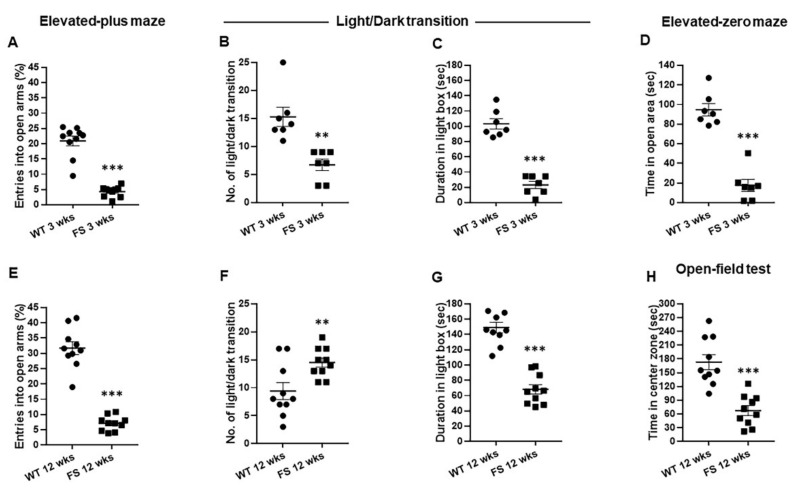
Increased anxiogenic behaviors in FS rats. Anxiety levels of FS 3 weeks rats were significantly enhanced in comparison with those in wild-type rats at 3 weeks. Percentage of entries into open arms in the elevated plus maze test (**A**). Number of light–dark transitions (**B**). Duration in the light box of the light–dark transition apparatus (**C**). Time in the light compartment in the zero maze test (**D**). Anxiety levels of FS 12 weeks rats were increased in comparison with those of wild-type rats at 12 weeks. Percentage of entries into the open arms in the elevated plus maze test (**E**). Light–dark transition number (**F**). Duration in the light box of the light–dark transition apparatus (**G**). Time in the central sector of the open field (**H**). Data are presented as means ± standard errors of the mean. ** *p* < 0.01, *** *p* < 0.001, two-tailed *t*-test (WT 3 weeks, *n* = 10; FS 3 weeks, *n* = 10; WT 12 weeks, *n* = 10; FS 12 weeks, *n* = 10).

**Figure 2 cells-11-03228-f002:**
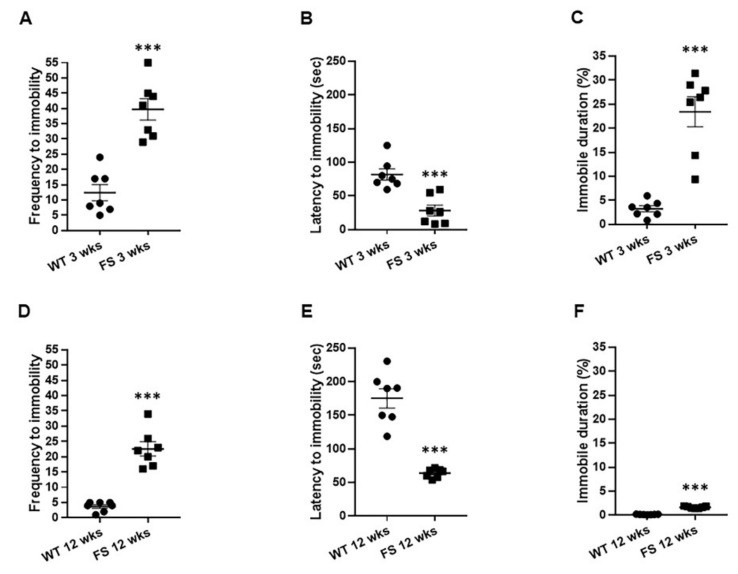
Increased depression-like behaviors in FS rats. Levels of depression-like behaviors of FS rats were significantly enhanced in comparison with those of wild-type littermates. Increased frequency of immobility in FS rats in the forced swim test (**A**,**D**). Decreased immobility latency in FS rats in the forced swim test (**B**,**E**). Increased percentage of immobility duration of FS rats in the forced swim test (**C**,**F**). Data are presented as means ± standard errors of the mean. *** *p* < 0.001, two-tailed *t*-test (WT 3 weeks, *n* = 7; FS 3 weeks, *n* = 7; WT 12 weeks, *n* = 7; FS 12 weeks, *n* = 7).

**Figure 3 cells-11-03228-f003:**
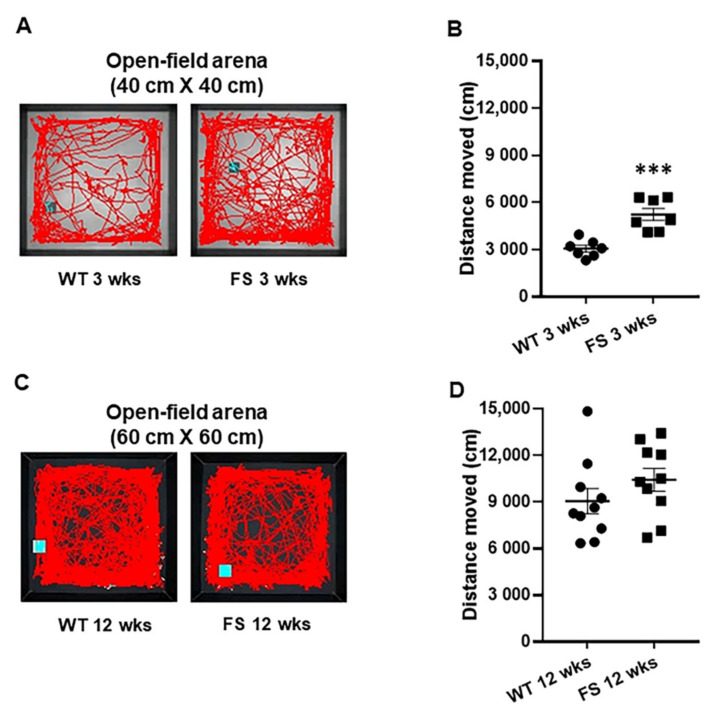
Increased levels of locomotion in FS juvenile rats, but not in FS 12 weeks rats. Representative movement traces of FS 3 weeks rats, FS 12 weeks rats, and wild-type littermates in open-field test (**A**,**C**). FS 3 weeks rats showed hyperlocomotor activities in comparison with wild-type rats after 3 weeks (**A**,**B**). FS 12 weeks rats and wild-type littermates showed similar levels of locomotor activities (**D**). Data are presented as means ± standard errors of the mean. *** *p* < 0.001, two-tailed *t*-test (WT 3 weeks, *n* = 7; FS 3 weeks, *n* = 7; WT 12 weeks, *n* = 10; FS 12 weeks, *n* = 10).

**Figure 4 cells-11-03228-f004:**
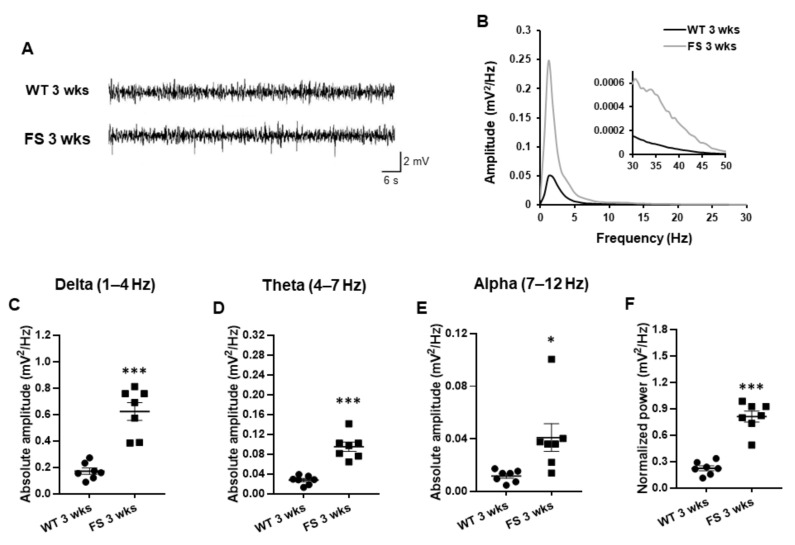
Increased delta and theta oscillations in the hippocampus of FS juvenile rats. Representative LFP traces in the hippocampus of FS 3 weeks rats and wild-type littermate rats (**A**). Power spectral analysis of FS 3 weeks rats revealed larger amplitude at approximately 1–5 Hz frequency than wild-type 3 weeks rats (**B**). Neuronal activities of FS 3 weeks rats at delta (**C**), theta (**D**), and alpha frequency waves (**E**) were markedly elevated in comparison with those of wild-type littermate rats. Representative power spectral analysis showed that averaged and normalized amplitude power of FS 3 weeks rats were stronger than those of wild-type rats after 3 weeks (**F**). Data are presented as means ± standard errors of the mean. * *p* < 0.05, *** *p* < 0.001, two-tailed *t*-test (WT 3 weeks, *n* = 7; FS 3 weeks, *n* = 7).

**Figure 5 cells-11-03228-f005:**
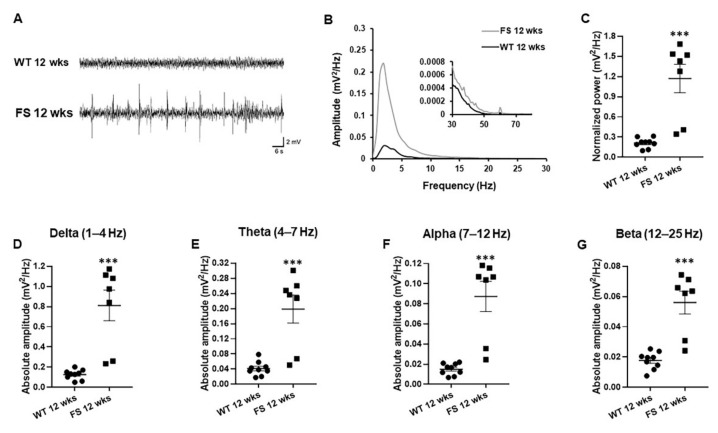
Increased delta, theta, alpha, and beta oscillations in the hippocampus of adult FS rats. Wild-type rats had generally representative LFP signals (**A**). Increased amplitude spikes of irregularly sharp wave and multispikes were examined in LFP signals of FS 12 weeks rats in comparison with littermate wild-type rats (**A**). Power spectral analysis showed that amplitude power at approximately 1–10 Hz frequency in FS 12 weeks rats was larger than that in wild-type rats after 12 weeks (**B**). Representative power spectral analysis showed that averaged and normalized amplitude power in FS 12 weeks rats was stronger than that in wild-type rats after 12 weeks (**C**). Neuronal activities at 1–25 Hz frequency, including delta (**D**), theta (**E**), alpha (**F**), and beta waves (**G**), in FS 12 weeks rats were increased in comparison with those in wild-type littermates. Data are presented as means ± standard errors of the mean. *** *p* < 0.001, two-tailed *t*-test (WT 12 weeks, *n* = 9; FS 12 weeks, *n* = 7).

## Data Availability

Not applicable.

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
