# Peer review of "Febrile Seizures Cause Depression and Anxiogenic Behaviors in Rats"

_cells, 2022, doi:10.3390/cells11203228_

Round 1

Reviewer 1 Report

The authors Yu et al., 2022 have given an account of comparative analysis of each emotional phenotype to improve the understanding of the behavioral sequelae following and/or during the epi- 75 leptogenic period caused by FS in SD rats. The flow of the manuscript from introduction to conclusion is well written and maintained the legacy of the paper. Although, there is some minor suggestion that should be incorporated for better readability of the paper before acceptance.

Minor comments:

1.     Title needs little amendment for impressive quality and readability.

2.     Research problem of the research is not clearly mentioned in the introduction part. Kindly elaborate on the novelty and research question in the last paragraph of the introduction. 

3.     Authors should add more literature in order to justify the research gap in their work as a lot of work on this topic is already reported.

4.     The quality of graphs should be slightly better. Kindly improve the quality if possible.

5.     Very little previous literature has been incorporated into the discussion. Kindly incorporate more studies for justifying your results.

6.     Kindly improve the quality of English and avoid grammatical errors.

7.     Maintain uniformity throughout the references.

8.     Ref 21 (Year is written in italics). Please correct.

9.     Ref 48 (Journal name should be written in italics and make the year bold).

10.  Authors should also consider some molecular work such as upregulation/downregulation of some factors responsible for depression and epilepsy in SD rats such as glial fibrillary proteins as well as gene profiling that would enhance the quality of the work if possible.

Author Response

Reviewer 1

The authors Yu et al., 2022 have given an account of comparative analysis of each emotional phenotype to improve the understanding of the behavioral sequelae following and/or during the epi- 75 leptogenic period caused by FS in SD rats. The flow of the manuscript from introduction to conclusion is well written and maintained the legacy of the paper. Although, there is some minor suggestion that should be incorporated for better readability of the paper before acceptance.

Minor comments:

  1. Title needs little amendment for impressive quality and readability.

Response:

Following reviewer’s suggestion, we amended title, “Febrile seizures cause depression and anxiogenic behaviors in rats”.

  1. Research problem of the research is not clearly mentioned in the introduction part. Kindly elaborate on the novelty and research question in the last paragraph of the introduction. 

Response:

With respect to reviewer’s comments, we re-write about last paragraph of introduction part as below (Page 2 Line 59 – 69):

Although the causal relationship between FS occurrence and emotional disorders [16-18] remains controversial, previous studies have clearly indicated that early develop-mental insults by FS can aggravate medial temporal lobe functions associated with facial emotional expression and recognition in human [19] and cause anxiety-related behaviors in adulthood of rodents [19]. However, most of prior studies were focused on epileptogenic activity of adulthood after early-life seizure impairment rather than primarily on behavioral changes. To the best of our knowledge, little data are available pertaining to the relationship between long-term effect of epilepsy and declined emotional phenotypes after FS. Thus, the objective of current study was to perform a comparative analysis of emotional phenotype using juvenile and adult rodents with FS to improve our understanding of behavioral sequelae of FS during the epileptogenic period and adulthood.

  1. Authors should add more literature in order to justify the research gap in their work as a lot of work on this topic is already reported.

Response:

With respect to reviewer’s comments, we inserted several references showing research gap in comparison with our current study in introduction and discussion part as below:

  1. Covolan L, Mello LEAM. Temporal profile of neuronal injury following pilocarpine or kainic acid-induced status epilepticus. Epilepsy Res 2000;39:133-52.
  2. Kornelsen, R.A.; Boon, F.; Leung, L.S.; Cain, D.P. The Effects of a Single Neonatally Induced Convulsion on Spatial Navigation, Locomotor Activity and Convulsion Susceptibility in the Adult Rat. Brain Res. 1996, 706, doi:10.1016/0006-8993(95)01245-1.
  3. Ettinger, A.B.; Weisbrot, D.M.; Nolan, E.E.; Gadow, K.D.; Vitale, S.A.; Andriola, M.R.; Lenn, N.J.; Novak, G.P.; Hermann, B.P. Symptoms of Depression and Anxiety in Pediatric Epilepsy Patients. Epilepsia 1998, 39:595-9, doi:10.1111/j.1528-1157.1998.tb01427.x.
  4. Castelhano, A.S.S.; Ramos, F.O.; Scorza, F.A.; Cysneiros, R.M. Early Life Seizures in Female Rats Lead to Anxiety-Related Behavior and Abnormal Social Behavior Characterized by Reduced Motivation to Novelty and Deficit in Social Discrimination. J. Neural Transm. 2015, 122(3):349, doi:10.1007/s00702-014-1291-2.
  5. Shin J, Gireesh G, Kim SW, et al. Phospholipase C β4 in the medial septum controls cholinergic theta oscillations and anxiety behaviors. J Neurosci 2009;29:15375-8.
  6. Hoeller, A.A.; Duzzioni, M.; Duarte, F.S.; Leme, L.R.; Costa, A.P.R.; Santos, E.C.D.S.; De Pieri, C.H.; Dos Santos, A.A.; Naime, A.A.; Farina, M.; et al. GABA-A Receptor Modulators Alter Emotionality and Hippocampal Theta Rhythm in an Animal Model of Long-Lasting Anxiety. Brain Res. 2013, 1532:21-31, doi:10.1016/j.brainres.2013.07.045.
  7. Adhikari, A.; Topiwala, M.A.; Gordon, J.A. Synchronized Activity between the Ventral Hippocampus and the Medial Prefrontal Cortex during Anxiety. Neuron 2010, 65:257-69, doi:10.1016/j.neuron.2009.12.002.
  8. Kobayashi, K.; Ikeda, Y.; Sakai, A.; Yamasaki, N.; Haneda, E.; Miyakawa, T.; Suzuki, H. Reversal of Hippocampal Neuronal Maturation by Serotonergic Antidepressants. Proc. Natl. Acad. Sci. U. S. A. 2010, 107:8434-9, doi:10.1073/pnas.0912690107.
  9. Di Gennaro, G.; Quarato, P.P.; Onorati, P.; Colazza, G.B.; Mari, F.; Grammaldo, L.G.; Ciccarelli, O.; Meldolesi, N.G.; Sebastiano, F.; Manfredi, M.; et al. Localizing Significance of Temporal Intermittent Rhythmic Delta Activity (TIRDA) in Drug-Resistant Focal Epilepsy. Clin. Neurophysiol. 2003, 114:70-8, doi:10.1016/S1388-2457(02)00332-2.

  1. The quality of graphs should be slightly better. Kindly improve the quality if possible.

Response:

Actually, all figures were submitted in accordance with the 600dpi resolution standard. Regarding reviewer’s comment, we will upload it separately with manuscript.

  1. Very little previous literature has been incorporated into the discussion. Kindly incorporate more studies for justifying your results.

Response:

Following reviewer’s comment, we added references showing supported the legitimacy of our results in the discussion part as below:

  1. Ettinger, A.B.; Weisbrot, D.M.; Nolan, E.E.; Gadow, K.D.; Vitale, S.A.; Andriola, M.R.; Lenn, N.J.; Novak, G.P.; Hermann, B.P. Symptoms of Depression and Anxiety in Pediatric Epilepsy Patients. Epilepsia 1998, 39:595-9, doi:10.1111/j.1528-1157.1998.tb01427.x.
  2. Castelhano, A.S.S.; Ramos, F.O.; Scorza, F.A.; Cysneiros, R.M. Early Life Seizures in Female Rats Lead to Anxiety-Related Behavior and Abnormal Social Behavior Characterized by Reduced Motivation to Novelty and Deficit in Social Discrimination. J. Neural Transm. 2015, 122(3):349, doi:10.1007/s00702-014-1291-2.
  3. Shin J, Gireesh G, Kim SW, et al. Phospholipase C β4 in the medial septum controls cholinergic theta oscillations and anxiety behaviors. J Neurosci 2009;29:15375-8.
  4. Hoeller, A.A.; Duzzioni, M.; Duarte, F.S.; Leme, L.R.; Costa, A.P.R.; Santos, E.C.D.S.; De Pieri, C.H.; Dos Santos, A.A.; Naime, A.A.; Farina, M.; et al. GABA-A Receptor Modulators Alter Emotionality and Hippocampal Theta Rhythm in an Animal Model of Long-Lasting Anxiety. Brain Res. 2013, 1532:21-31, doi:10.1016/j.brainres.2013.07.045.
  5. Adhikari, A.; Topiwala, M.A.; Gordon, J.A. Synchronized Activity between the Ventral Hippocampus and the Medial Prefrontal Cortex during Anxiety. Neuron 2010, 65:257-69, doi:10.1016/j.neuron.2009.12.002.
  6. Kobayashi, K.; Ikeda, Y.; Sakai, A.; Yamasaki, N.; Haneda, E.; Miyakawa, T.; Suzuki, H. Reversal of Hippocampal Neuronal Maturation by Serotonergic Antidepressants. Proc. Natl. Acad. Sci. U. S. A. 2010, 107:8434-9, doi:10.1073/pnas.0912690107.
  7. Di Gennaro, G.; Quarato, P.P.; Onorati, P.; Colazza, G.B.; Mari, F.; Grammaldo, L.G.; Ciccarelli, O.; Meldolesi, N.G.; Sebastiano, F.; Manfredi, M.; et al. Localizing Significance of Temporal Intermittent Rhythmic Delta Activity (TIRDA) in Drug-Resistant Focal Epilepsy. Clin. Neurophysiol. 2003, 114:70-8, doi:10.1016/S1388-2457(02)00332-2.

  1. Kindly improve the quality of English and avoid grammatical errors.

Response:

Following the reviewer’s comments, we re-checked and corrected for the grammatical errors in revised manuscript with the professional correctional institution.

  1. Maintain uniformity throughout the references.

Response:

We are sorry for our mistake, and deleted word template of reference part.

  1. Ref 21 (Year is written in italics). Please correct.

Response:

Ref21 has been rewritten in bold (Page 11 Line 451).

  1. Ref 48 (Journal name should be written in italics and make the year bold).

Response:

Ref 48 has been rewritten following reviewer’s comment (Page 13 Line 532).

  1. Authors should also consider some molecular work such as upregulation/downregulation of some factors responsible for depression and epilepsy in SD rats such as glial fibrillary proteins as well as gene profiling that would enhance the quality of the work if possible.

Response:

Thank you for good comments. In the current study, we would like to suggest causal relationship between FS occurrence and emotional disorders, such as depression and anxiety, using behavioral results in rodent. In case of depression and epilepsy-related molecular results, with respect to inflammation responses according to FS through cell markers for astrogliosis/microgliosis experiments are already underway to present, however, it will be submitted for a follow-up paper. We respectfully ask reviewers to understand our intention and plans.

Reviewer 2 Report

In my opinion, the manuscript entitled “Anxiogenic and depression-like behaviors following febrile seizures in rat”  is generally very well organized and well-written. The chosen subject is interesting and important. I have only several minor suggestions:

* In the introduction section the Authors should mention more about the carried out study, its hypotheses, measured parameters.

* In the Results section, % of time spent in open arms of the EPM should be presented.

*The Conclusion section should be changed. Conclusions are too far-reaching. ADHD, schizophrenia, and autism were not the subject of the presented project.

Author Response

Reviewer 2

In my opinion, the manuscript entitled “Anxiogenic and depression-like behaviors following febrile seizures in rat” is generally very well organized and well-written. The chosen subject is interesting and important. I have only several minor suggestions:

  1. In the introduction section the Authors should mention more about the carried out study, its hypotheses, measured parameters.

Response:

With respect to reviewer’s comments, we supplement references and rewrite of introduction part as below (Page 2 Line 54 – 69).

However, in SE models, neuronal loss is not restricted to the hippocampus. It also occurs in the parahippocampal cortical regions, the thalamus, the endopiriform cortex, and the amygdala [10 – 13]. Among these areas, amygdala is involved in cognitive deficits and emotional disorders in humans and other animals [14,15].

Although the causal relationship between FS occurrence and emotional disorders [16-18] remains controversial, previous studies have clearly indicated that early develop-mental insults by FS can aggravate medial temporal lobe functions associated with facial emotional expression and recognition in human [19] and cause anxiety-related behaviors in adulthood of rodents [19]. However, most of prior studies were focused on epileptogenic activity of adulthood after early-life seizure impairment rather than primarily on behavioral changes. To the best of our knowledge, little data are available pertaining to the relationship between long-term effect of epilepsy and declined emotional phenotypes after FS. Thus, the objective of current study was to perform a comparative analysis of emotion-al phenotype using juvenile and adult rodents with FS to improve our understanding of behavioural sequelae of FS during the epileptogenic period and adulthood.

  1. In the Results section, % of time spent in open arms of the EPM should be presented.

Response:

In our results, Figure 1A and E represent % of time spend in open arms in the elevated-plus maze. We are sorry for your confusing, and y-axis title of the graph has been rewritten (Page 5, Figure 1A and 1E).

  1. The Conclusion section should be changed. Conclusions are too far-reaching. ADHD, schizophrenia, and autism were not the subject of the presented project.

Response:

With respect to reviewer’s comments, we re-write our conclusions part of manuscript as below (Page 10 Line 373-378).

In conclusion, our findings suggest that functional imbalances of GABAergic inhibition and abnormalities of unnatural neurogenesis in hippocampal dentate granule cells might lead to time-dependent emotional phenotypes and specific brain waves. In addition, changes of neural oscillations in the hippocampus might be related to cognitive dysfunctions and anxiety disorders, which might be useful as neural oscillatory markers representing depression and emotional disorders.